# POSTERIOR-GUIDED VISUAL TOKEN PRUNING IN VISION–LANGUAGE MODELS

## ABSTRACT

Vision-language models (VLMs) with dynamic resolution vision encoders achieve strong performance, but face significant efficiency challenges due to long input sequences. A common approach is to assess the importance of tokens and prune those that are less informative. Recent methods utilizing a small VLM to provide the importance map of visual tokens have outperformed existing rule-based and similarity-driven pruning approaches, particularly under high pruning ratios. However, directly using the small VLM remains unreliable, as it aggregates cross-attention weights between all the generated answer tokens of the small VLM and the visual inputs to form an importance map, which can lead to noisy guidance if the generated answer is incorrect. To address this, we invert the approach by having it detect non-informative visual tokens according to the user's input query. By adding a learnable information bottleneck in the small VLM, we can approximate the posterior distribution of non-important visual tokens. This enables the small model to highlight broad informative regions, allowing the large VLM to retain its reasoning capacity with improved efficiency. Extensive experiments on eight benchmarks demonstrate the effectiveness of our approach. With only 5% of visual tokens retained, the large VLM preserves 95% of its original performance, outperforming the state of the art by 8%.

## 1 INTRODUCTION

Vision–language models (VLMs) have demonstrated remarkable progress across a wide range of visual tasks, yet their deployment remains hindered by high computational costs. A key source of inefficiency arises from the large number of visual tokens produced by dynamic resolution encoders, which significantly increases sequence length and burdens downstream reasoning Bai et al. (2025). However, the visual input has high redundancy and sparsity as a generation condition in VLM. Therefore, token pruning (Ye et al., 2025; Ma et al., 2024; Li et al., 2024b; Yang et al., 2025; Bolya et al., 2023) has emerged as a promising strategy for improving efficiency by discarding less informative visual tokens.

Despite recent progress, existing pruning approaches suffer from fundamental limitations. For example, FastV (Chen et al., 2024a) assumes that cross-attention from the first generated token provides a reliable signal of token importance. In practice, however, this assumption often breaks down, leading to unstable pruning decisions. More recent work, SGP (Zhao et al., 2025) aggregates attention scores across all generated tokens from a small-VLM to construct importance maps, which are then used to guide pruning for a larger VLM with the same architecture. While this strategy yields improvements under high pruning ratios, its pruning guidance is heavily dependent on the answering ability of the small model. This reliance restricts generalization to complex instructions that have higher visual dependency. As shown in Fig. 1a, when the small-VLM lacks the prior knowledge to answer a given query, the importance map it produces becomes ineffective, resulting in noisy token retention and impairing the large-VLM's reasoning capacity.

To address this, we invert the paradigm: rather than asking a small model with limited ability to identify the most important visual tokens and forcing the large model to follow, we instead train the small model to approximate the distribution of non-informative tokens. As shown in Fig. 1b, the small-VLM learns to map low-informative visual tokens conditioned on the user input to a learnable prior via a bottleneck module. Implicitly, the tokens far from the prior are treated as important

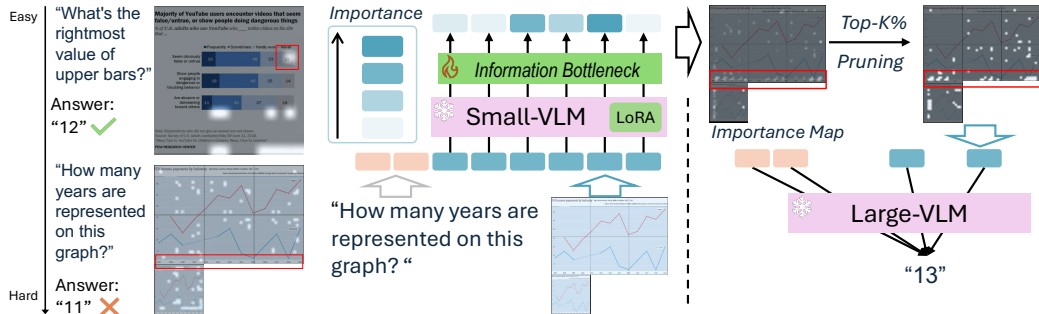

(a) Pruning guidance by SGP.  (b) Inference pipeline of our proposed pruning method (PGP).

Figure 1: **(a)** SGP utilizes a pre-trained VLM for the importance map prediction, but failed to provide helpful pruning guidance due to its answer-driven mechanism. **(b)** We fine-tuned an information bottleneck module to map the output visual embeddings from a small-VLM to a latent variable, which are used to compute the importance of each visual token given the provided text prompt. The pruning guidance is more helpful than SGP after top-$K\%$ pruning for the large-VLM.

ones, which will be passed to the large-VLM for reasoning and final answering. In this way, the small-VLM can highlight broader regions of informative content than the attention-based guidance, making the pruning guidance less deterministic but more inspirational and auxiliary, and enabling the large-VLM to retain its full reasoning capability with better efficiency. Overall, we posit that effective pruning requires moving beyond answer-driven heuristics and instead estimating token importance through a principled probabilistic framework. To this end, we introduce **P**osterior-**G**uided token **P**runing (**PGP**), which formulates token importance estimation as an amortized variational inference problem. Specifically, we fine-tune a small-VLM to act as a latent variable sampler, and we can approximate the distribution over each visual token's contribution to the downstream task by parameterizing each latent visual token as a channel-wise Gaussian. Intuitively, this posterior-driven formulation yields pruning guidance that is both query- and answer-aware, ensuring that the retained visual tokens are more informative for reasoning. By computing the KL-divergence between the predicted posterior and the prior distribution as an importance score, PGP leverages the small-VLM to provide a robust and transferable estimate of visual relevance.

PGP is also practical for inference. Prior methods compute importance inside the large VLM's decoder or require full decoding to an end-of-sequence token, incurring significant overhead. In contrast, PGP produces guidance in a single forward pass of the small VLM, substantially reducing the large model's FLOPs and memory without sacrificing accuracy. Furthermore, our method works well with optimized libraries, such as FlashAttention (Dao, 2023), since it does not require explicitly outputting all the attention weights, unlike SGP and FastV. Extensive experiments demonstrate that PGP achieves state-of-the-art trade-offs between efficiency and performance. Specifically, PGP retains up to 95% of the original accuracy while utilizing only 5% of the visual tokens, resulting in a 40% reduction in computational cost, and consistently outperforms previous SOTA methods by 7% across eight benchmarks.

## 2 PRELIMINARIES

**Vision–Language Models.** General-purpose vision–language models (VLMs) are typically instantiated as causal large language models (LLMs) conditioned on visual inputs. To strengthen visual understanding, recent families such as LLaVA (Li et al., 2024a; Sun et al., 2024), QwenVL (Bai et al., 2025; Wang et al., 2024), and InternVL (Chen et al., 2024d;e) adopt dynamic visual encoders. A high-resolution image is tiled into multiple crops; each crop is passed through a ViT to produce a fixed-length sequence of patch embeddings. Let $X \in \mathbb{R}^{n \times d}$ denote textual embeddings and $V \in \mathbb{R}^{m \times d}$ denote visual embeddings, both projected into the LLM's $d$-dimensional space, and let $N = m + n$. The LLM consumes the concatenated sequence $[V; X] \in \mathbb{R}^{N \times d}$ and outputs contextualized hidden states $[V', X']$, in which visual representations mutually inform text information.

At inference time (pre-fill), the VLM forms a single causal sequence over visual and textual tokens. With an appropriate attention mask, each visual token can attend to previously seen text (e.g., user instructions and system prompts), allowing the visual features to be shaped by the query context.

**Small-VLM–Guided Visual Token Pruning.** While longer visual-token sequences can improve fine-grained perception, many tokens are empirically redundant. Selecting only the most informative tokens trades off efficiency and accuracy (Zhang et al., 2024b; Alvar et al., 2025; Chen et al., 2024a). Small-VLM–Guided Pruning (SGP) (Zhao et al., 2025) addresses this by deriving a token-level *importance map* $s \in \mathbb{R}^m$ from a compact VLM. Given $s$, the top-$k\%$ visual tokens are retained and the remainder are hard-pruned before the visual input is forwarded to the answer predictor (i.e., the large VLM).

# 3 Posterior-Guided Visual Token Pruning via Variational Inference

## 3.1 Variational Information Bottleneck in VLMs

A central challenge in visual token pruning is determining the relative importance of each token without introducing significant computational overhead. We propose a *token-wise variational information bottleneck* framework, which treats each visual token as a stochastic latent variable and leverages a Kullback–Leibler (KL) divergence to quantify its information contribution.

Formally, given a set of visual embeddings $\boldsymbol{V}' = \{\boldsymbol{V}'_1, \ldots, \boldsymbol{V}'_m\}$ after the small-VLM forward pass, we map them into latent variables $\boldsymbol{Z} = \{\boldsymbol{Z}_1, \ldots, \boldsymbol{Z}_m\}$, where each $\boldsymbol{Z}_i \sim Q_\theta(\boldsymbol{Z}_i \mid \boldsymbol{V}'_i)$ is the latent representation of the $i^{th}$ visual token. The token-wise latent variable is represented by a Gaussian distribution as:

$$Q_\theta(\boldsymbol{Z}_i \mid \boldsymbol{V}'_i) = \mathcal{N}\big(\mu_\theta(\boldsymbol{V}'_i), \sigma^2_\theta(\boldsymbol{V}'_i)\big), \tag{1}$$

where $\mu_\theta(\boldsymbol{V}'_i), \sigma^2_\theta(\boldsymbol{V}'_i) \in \mathbb{R}^d$ are predicted by a projection layer parameterized by $\theta$. Specifically, in sequence-to-sequence LLM with causal attention, $\boldsymbol{V}'$ can naturally fuse the prior query information by cross-attention over $\boldsymbol{X}$ as shown in Fig. 2. In this way, by conditioning on $\boldsymbol{V}'$, the latent variable prediction is implicitly conditioned on both query and visual information.

To encourage disentanglement across channels, we adopt a learnable prior distribution $P(\boldsymbol{z}) = \mathcal{N}(\mu_p, \sigma^2_p)$, where $\mu_p, \sigma^2_p \in \mathbb{R}^d$ are per-channel learnable mean and variance that are shared over the whole training data space. This design allows certain latent dimensions to carry more informative content while encouraging redundancy reduction in less important dimensions.

To this end, the KL divergence between the approximate posterior and prior,

$$D_{\mathrm{KL}}(Q_\theta(\boldsymbol{Z}_i \mid \boldsymbol{V}'_i) \| P(\boldsymbol{z})) := \frac{1}{d} \sum_{j=1}^{d} D_{\mathrm{KL}}(Q_\theta(\boldsymbol{Z}_i^{(j)} \mid \boldsymbol{V}_i^{'(j)}) \| P(\boldsymbol{z}^{(j)})), \tag{2}$$

representing the average amount of channel-wise information the $i^{th}$ visual token contributes beyond the prior belief with $d$ dimension. Intuitively, tokens that deviate strongly from the prior carry higher task-relevant information, while tokens with near-prior distributions contribute little. Thus, the KL divergence naturally serves as a token-wise *importance score*, forming the basis for pruning guidance at inference time.

Instead of directly predicting the posterior mean of the $i^{th}$ visual token (i.e., $\mu_\theta(\boldsymbol{V}'_i)$) using a projection layer, we introduce a channel-wise gating mechanism to enhance the expressivity of the posterior mean as:

$$\mu_\theta(\boldsymbol{V}'_i) = \sigma\big(I_\theta(\boldsymbol{V}'_i)\big) \odot (\boldsymbol{V}'_i - \mu_p) + \mu_p, \tag{3}$$

where $I_\theta(\boldsymbol{V}'_i)$ is a learned channel-wise importance gate, $\sigma(*)$ is a sigmoid function, and $\odot$ denotes element-wise multiplication. This mechanism enables the model to independently modulate how much information each channel contributes to the posterior within a bound, since $0 < \sigma(*) < 1$. So the gate upper-bounds how far the posterior mean can wander from the prior, directly capping the KL explosion and stabilizing optimization. However, a free mean projection can push $\mu_\theta(\boldsymbol{V}'_i)$ arbitrarily far, making the KL term volatile.

Figure 2: The overview of our training pipeline. For each data sample $X, V, Y \sim \mathcal{D}$, we call small-VLM (i.e., $\pi_\phi$) twice. $1^{st}$ **forward:** Input $X$ and $V$, and the output $V'$ are mapped to latent space using $Q_\theta(*)$, computing KL divergence with a shared prior. $2^n$ **forward:** compute the cross-entropy between the predicted answer $\pi_\theta(Y|X, Z)$ and the ground truth $Y$.

## 3.2 RECONSTRUCTION OBJECTIVE

Our training objective extends the classical variational information bottleneck (Alemi et al., 2016; Tishby et al., 2000) to the token level, which can be easily applied to token prediction within LLM. Overall, the goal of our objective function is twofold: ❶ to ensure the accurate reconstruction of the target output conditioned on both the query and the latent visual tokens, and ❷ to penalize redundant tokens by compressing their representations towards the prior.

While representing each visual token by a continuous Gaussian distribution makes the optimization gradient intractable. Following the **reparameterization trick**, we "reparameterize" the distribution $Q_\theta(Z_i \mid V_i')$ as,

$$Z_i \sim Q_\theta(Z_i \mid V_i') = \mathcal{N}(\mu_\theta(V_i'), \sigma_\theta^2(V_i')) \Rightarrow Z_i = \mu_\theta(V_i') + \sigma_\theta(V_i') \cdot \epsilon, \quad (4)$$

where $\epsilon \in \mathcal{N}(0, I)$ is an auxiliary noise variable. Conditioning on the original query $X$ and the reparameterized latent visual tokens $Z = \{Z_1, \cdots, Z_m\}$, the reconstruction loss aims to maximize the expectation of the log-likelihood of the final answer $Y$. To this end, the overall loss is:

$$\mathcal{L} = \underbrace{\mathbb{E}_{X, Y \sim \mathcal{D}, Z}\left[\log \pi_\phi(Y \mid X, Z)\right]}_{\text{Reconstruction loss}} - \frac{\beta}{m}\sum_{i=1}^{m}\underbrace{D_{\text{KL}}\left(Q_\theta(Z_i \mid V_i')\|P(z)\right)}_{\text{Token-wise KL penalty}}, \quad (5)$$

where $\pi_\phi$ is the conditional likelihood modeled by the VLM with parameters $\phi$, and $\beta$ is a trade-off hyperparameter.

The reconstruction loss ensures the latent tokens preserve sufficient information for accurate answer prediction, while the KL term regularizes each token against the prior. By applying this penalty at the token level rather than the sequence level, we achieve two key benefits: 1. **Granular importance estimation.** Each token's KL divergence reflects its marginal utility for the downstream task, enabling fine-grained pruning decisions. 2. **Adaptive compression.** The learnable per-channel prior allows the model to automatically retain highly informative latent dimensions while suppressing redundancy.

In summary, Eq. 5 enforces a principled token-wise trade-off between predictive sufficiency and compression. The resulting KL-based importance scores can be directly employed as a pruning criterion, yielding both interpretability and computational efficiency.

## 3.3 POSTERIOR-GUIDED VISUAL TOKEN PRUNING

At inference time, we first pass the concatenated sequence of text and visual tokens (i.e., $XV$) to a small-VLM (S-VLM). Then, we extract $V'$ from the last hidden-states of the output sequence $(X'V')$, where $V'$ are new visual embeddings containing query information. Next, we approximate the latent variable of each visual token (i.e., $Z_i \sim Q_\theta(Z_i \mid V_i')$), then compute the importance score for each visual token (i.e., $s \in \mathbb{R}^m$), which are used to guide the token pruning in large-VLM (i.e., L-VLM). This work computes importance map $s = \{D_{\text{KL}}(Q_\theta(Z_1 \mid V_1')\|P(z)), \cdots, D_{\text{KL}}(Q_\theta(Z_m \mid$

$V'_m)\|P(z))\}$, where the posterior prediction layers parameterized by $\theta$ and the priors $P(z)$ were optimized in the training process. To be noticed, S-VLM should share the same model architecture as the L-VLM for consistent visual encoding. Following SGP (Zhao et al., 2025), we retain the top-$k\%$ of visual tokens by the ranked $vs$. Then, we perform hard pruning on $V$ and the pre-computed position embeddings. In this way, the remaining visual tokens still contain the original spatial information from the whole visual input.

# 4 EXPERIMENTAL RESULTS

## 4.1 IMPLEMENTATION

**Training recipe.** This work utilizes the InternVL family models for experiments, since the model architecture was designed to take the full visual sequence without compression, which clearly demonstrates the effectiveness of our method. For the small-VLM (i.e., $\pi_\phi(*)$), we initialize and fix the model parameters from the pretrained InternVL2.5-1B (Chen et al., 2024e), followed by a learnable light-weight projection module (i.e., $Q_\theta(*)$), which consists of two MLP layers, and two learnable embeddings (prior mean $\mu_p$ and variance $\sigma_p^2$). To alleviate the domain shift between $\pi_\phi(Y|X, V)$ and $\pi_\phi(Y|X, Z)$ during training, we fine-tune the small-VLM using Eq. 5 with LoRA for one epoch. Please refer to the supplementary material for our detailed hyperparameters. For better generalizability, we follow the training data used in InternVL, which is a mixture of single-image instruction data proposed by ShareGPT-4V (Chen et al., 2024b), LLaVA'(Li et al., 2024a), and DVQA (Kafle et al., 2018), etc.

**Benchmarks and baseline pruning methods.** We consider the pruning guidance as a general-purpose assistant, and this work focuses on single-image tasks. For OCR and chart understanding, we utilize TextVQA (Singh et al., 2019) and ChartQA (Masry et al., 2022). To validate the capabilities in real-world scenarios with open-form instructions, we utilize MMStar (Chen et al., 2024c) and RealWorldQA. Besides general visual understanding, visual perception evaluates the model's reasoning ability, so we adopt MME (Fu et al., 2023), MMBench (Liu et al., 2024), MM-Vet (Yu et al., 2024), and GQA (Hudson & Manning, 2019). For a fair comparison with other pruning methods, we primarily report the baseline results of our own implementation and utilize lmms-eval (Zhang et al., 2024a) for consistency in the test setting. All our results are reported with greedy sampling and zero-shot prediction. We carefully choose four pruning methods to compare with ours, where ToME (Bolya et al., 2023) solely focuses on reducing visual redundancy in the vision encoder, FastV (Chen et al., 2024a) progressively reduces the number of visual tokens in the LLM forward process based on attention weights, and SGP (Zhao et al., 2025) utilizes a small-VLM to provide pruning guidance, which is similar to our approach.

## 4.2 PGP PROVIDES RELIABLE PRUNING GUIDANCE

PGP works as a soft visual cue detector with reasoning capability, rather than providing only direct, point-wise pruning guidance. Existing approaches, such as FastV and SGP, adopt answer-driven pruning strategies, where the retained visual tokens are either directly tied to the predicted answer or consist largely of irrelevant noise. This strong reliance on a VLM's prior knowledge inherently limits both generalization and robustness. In contrast, by assigning higher importance scores to a broader set of potentially informative visual tokens (see Fig. 3), PGP preserves contextual cues that are critical for downstream reasoning. This enables the large VLM to perform more precise and fine-grained inference, ensuring both reliability and trustworthiness in pruning decisions. As illustrated in Fig. 4, PGP consistently identifies

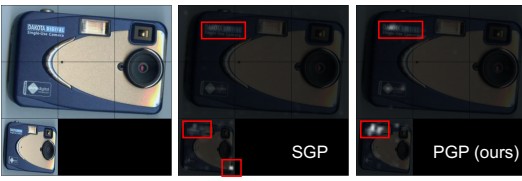

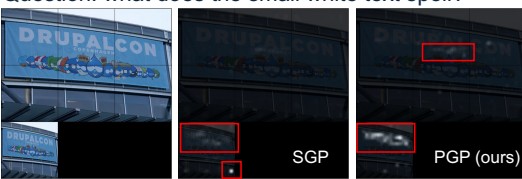

Figure 3: Visualization of visual token importance map proposed by SGP and PGP (ours).

Table 1: Comparison of InternVL2-26B with different visual token pruning methods. After obtaining the importance map using different methods, including FastV, SGP, and PGP, we retain the top-$k\%$ (i.e., token ratio) of all input visual tokens and execute hard pruning at the $L^{th}$ decoder layer of InternVL2-26B. † are results based on our reproduced experiments. The best results are **bold**.

| Method | Token ratio | L | TextVQA | ChartQA | GQA | MMStar | MMBench | MM-Vet | MME | RealWorldQA | Score ratio ↑ |
|---|---|---|---|---|---|---|---|---|---|---|---|
| | | | val | All | test-dev | test | en-dev | test | test | test | |
| InternVL2-26B | 100% | - | 82.45 | 84.92 | 64.89 | 60.08 | 83.46 | 64.00 | 2270 | 67.58 | 100.00% |
| ToMe | 20% | 9 | 75.74 | 62.44 | 63.61 | - | **81.82** | 52.50 | 2178 | - | 94.88% |
| FastV† | 20% | 9 | 75.62 | 71.68 | 61.20 | 53.01 | 78.31 | 45.00 | 2140 | 63.27 | 93.18% |
| SGP† | 20% | 9 | **81.97** | 81.68 | **64.62** | 56.77 | 80.76 | **62.34** | 2258 | **67.50** | 99.15% |
| PGP (ours) | 20% | 9 | 81.48 | **82.60** | 64.56 | **57.46** | 80.58 | 61.01 | **2271** | 66.14 | **99.55%** |
| FastV† | 20% | 0 | 73.42 | 67.32 | 60.68 | 50.55 | 78.26 | 52.66 | 2110 | 60.26 | 90.03% |
| SGP† | 20% | 0 | 81.14 | 80.92 | 64.70 | **56.97** | **80.50** | **61.33** | 2252 | **67.90** | 98.49% |
| PGP (ours) | 20% | 0 | **81.28** | **82.36** | **64.86** | 56.45 | 79.98 | 60.32 | **2263** | 66.54 | **99.19%** |
| ToMe | 5% | 2 | 51.69 | 28.60 | 57.52 | - | 73.09 | 37.70 | 1933 | - | 82.33% |
| FastV† | 5% | 2 | 43.84 | 26.10 | 44.90 | 32.65 | 62.33 | 31.60 | 1799 | 44.05 | 75.05% |
| SGP† | 5% | 2 | 78.70 | 71.08 | 62.04 | 50.92 | 73.71 | 49.82 | 2007 | 64.84 | 88.50% |
| PGP (ours) | 5% | 2 | **79.24** | **71.12** | **63.52** | **53.10** | **77.58** | 50.83 | **2189** | **65.62** | **95.41%** |
| FastV† | 5% | 0 | 20.06 | 24.64 | 43.41 | 32.65 | 36.94 | 21.74 | 1418 | 44.05 | 59.10% |
| SGP† | 5% | 0 | 78.77 | 70.68 | 62.08 | 50.62 | 73.28 | 50.23 | 2028 | **65.10** | 89.25% |
| PGP (ours) | 5% | 0 | **79.04** | **70.96** | **63.53** | **52.49** | **77.23** | **51.42** | **2190** | 66.01 | **95.44%** |

semantically meaningful and question-related visual cues, whereas SGP primarily localizes tokens directly tied to the predicted answer. We quantify the difficulty of human instructions by their degree of visual dependency, with more complex queries requiring a larger set of relevant visual tokens for accurate reasoning. Under this characterization, we observe that pruning decisions guided by SGP enable the large VLM to handle relatively simple queries but result in substantial performance degradation on visually demanding ones.

Quantitative results in Table 1 further validate this advantage. With only 20% of visual tokens retained, PGP and SGP preserve 99.4% and 98.82% of the original performance (measured by the score ratio), respectively, indicating less than a 1% drop. However, under more aggressive pruning, the performance gap widens significantly: with just 5% of tokens, PGP still maintains 95.4% of the full performance, while SGP and FastV degrade to 88.9% and 67.1%, respectively. Fig. 5 also shows that pruning guided by PGP yields more stable performance than competing methods. Together with the qualitative examples in Fig. 4, these results support our main hypothesis that preserving highly informative tokens is more effective than relying solely on answer-related ones.

### 4.3 ONE CAN SERVE MANY

We study whether a single fine-tuned small VLM can guide pruning for larger models that share the same architecture. Reusing the guide model from Table 1 (InternVL2.5-1B), we prune InternVL2-8B with $L = 0$ and report results in Table 2. At a moderate retention ($R = 20\%$), both SGP and PGP remain close to the full-token baseline (normalized score

Table 2: Performance comparison of InternVL2-8B with different pruning methods including SGP and PGP.

| Method | R | GQA | MMStar | MMBench | RealWorldQA | Score % |
|---|---|---|---|---|---|---|
| InternVL2-8B | 100% | 62.70 | 59.11 | 81.90 | 65.10 | 100% |
| SGP | 20% | **62.59** | 56.37 | **80.67** | **64.58** | **98.29%** |
| PGP (ours) | 20% | 62.54 | **56.93** | 79.64 | 63.14 | 97.56% |
| SGP | 5% | **59.95** | 50.37 | 71.22 | 61.31 | 90.34% |
| PGP (ours) | 5% | 58.47 | **53.34** | **76.46** | **62.48** | **94.03%** |

98.29% vs. 97.56%). PGP is slightly better on MMStar ($+0.56$), while SGP is marginally higher on MMBench ($-1.03$) and RealWorldQA ($-1.44$), with parity on GQA ($-0.05$). Under aggressive pruning ($R = 5\%$), PGP clearly surpasses SGP at the same retention: 94.03% vs. 90.34% ($+3.69$). Gains are largest on MMBench ($+5.24$) and MMStar ($+2.97$), with a smaller improvement on RealWorldQA ($+1.17$) and a modest drop on GQA ($-1.48$). These findings are consistent with Table 1 and indicate that the amortized posterior learned by the small VLM transfers reliably across model scales within the InternVL family. In practice, we highlight the $R = 5\%$ regime for its superior performance–efficiency trade-off.

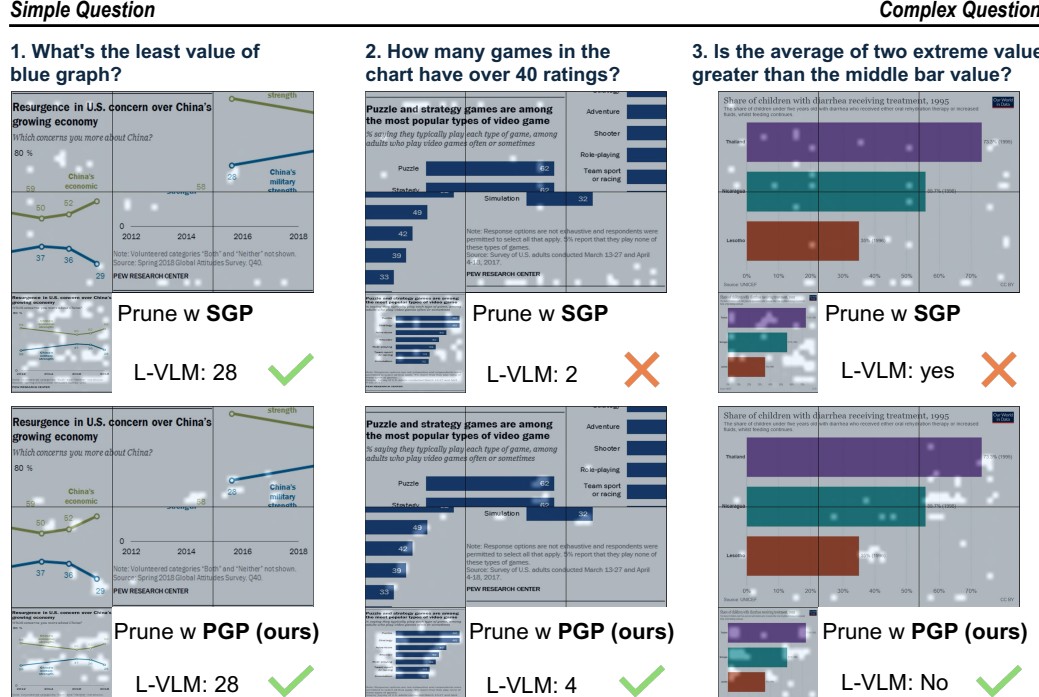

Figure 4: Comparison of the same large-VLM (L-VLM) with different pruning methods. For each visual input, we highlight the top-5% of all the visual tokens based on the importance map predicted by SGP and PGP. **Upper three:** SGP provides answer-driven pruning guidance, impacting the large-VLM's performance. **Lower three:** PGP provides posterior-driven guidance, where the retained visual tokens are high query and answer relevance, allowing the L-VLM to perform sufficient visual understanding before answering.

| Method | R | L | S-F | L-F | FLOPs % ↓ | Score % ↑ |
|---|---|---|---|---|---|---|
| InternVL2-26B | 100% | - | - | 117.7T | 100.0% | 100% |
| SGP | 20% | 9 | | 81.4T | 81.5% | 99.15% |
| | 5% | 2 | 14.5T | 67.5T | 69.7% | 88.50% |
| | 5% | 0 | | 65.4T | 67.9% | 89.25% |
| PGP (ours) | 20% | 9 | | 83.4T | 74.6% | 99.55% |
| | 5% | 2 | 4.7T | 69.3T | 62.9% | 95.41% |
| | 5% | 0 | | 67.3T | 61.2% | 95.44% |

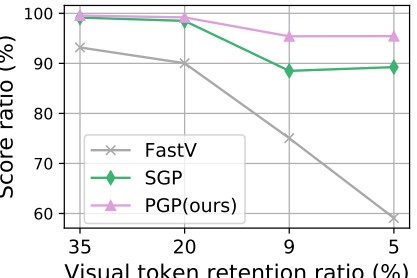

Table 3: Performance and FLOPs of different pruning methods. We prune $100 - R(\%)$ of visual tokens at $L^{th}$ decoder layer. S-F and L-F indicate the inference FLOPs of the small- and large-VLM.

Figure 5: Performance–efficiency curve. PGP demonstrates greater stability under progressively fewer visual tokens, preserving accuracy more effectively.

## 4.4 IMPROVING EFFICIENCY WITH LESS PERFORMANCE-DROPS

Although PGP utilizes a small VLM to guide pruning for the large VLM, our experiments demonstrate that even with an auxiliary model, the total computational cost (FLOPs) of the system remains reduced compared to using the large VLM alone. Table 3 reports the average FLOPs over 100 samples. PGP achieves lower FLOPs in the small VLM because it requires only a single forward pass to generate pruning guidance, whereas SGP requires a complete answer generation. While PGP incurs slightly higher FLOPs in the large VLM—due to the growth of inference cost with generated token length—it remains more efficient overall. On average, PGP reduces FLOPs by 34% with only a 3.2% performance drop, demonstrating a favorable trade-off for practical deployment.

Table 4: Ablation study of major components in our proposed objective function for training the information bottleneck. We report the results of InternVL2-26B after pruning guided by our small-VLM using PGP with $R = 5\%$ and $L = 0$ for all the models. $\beta$ is the weight of the KL penalty and $f(*)$ is the gateway activation for posterior mean approximation.

| $\beta$ | $f(*)$ | ChartQA | GQA | MMStar | MMBench | TextVQA | MM-Vet | RealWorldQA | Score % ↑ |
|---|---|---|---|---|---|---|---|---|---|
| - | - | 84.92 | 64.89 | 60.08 | 83.46 | 82.45 | 64.00 | 67.58 | 100% |
| 0.5 | exp | 69.92 | 63.34 | 52.25 | 77.15 | 78.28 | 49.63 | 65.23 | 89.83% |
| 0.5 | $\sigma$ | 70.28 | 63.77 | 52.58 | 75.77 | 78.72 | 53.30 | 66.27 | 90.80% |
| $\tau(0.2, 0.5)$ | $\sigma$ | 70.96 | 65.53 | 52.49 | 77.23 | 79.04 | 51.42 | 66.01 | 91.19% |

## 4.5 ABLATION STUDY

**Effect of channel-wise gating activation.** We compare the exponential and sigmoid ($\sigma$) functions as channel-wise gating activations. While both methods achieve non-trivial compression, sigmoid consistently outperforms exponential gating across benchmarks (90.80% vs. 89.83% overall score). This suggests that sigmoid provides a more independent and stable allocation of importance weights across channels, as its normalized scaling is from $(0, 1)$. This reduces the gradient spikes and variance in the KL term, preventing the over-amplification of individual channels.

**Training with adaptive KL weight.** Our objective in Eq. 5 has a KL penalty term, aiming to compress the information of non-query and non-answer correlated visual tokens. A fixed coefficient $\beta$ can, however, lead to suboptimal trade-offs across different training stages. In the early phase of optimization, a strong KL regularization may suppress informative tokens prematurely, hindering reconstruction fidelity. Conversely, a weak regularization in later stages can result in redundant token retention and slow convergence.

To mitigate this issue, we adopt an *adaptive KL weighting* strategy. Concretely, we introduce a schedule $\beta(s) = \tau_{max} - (\tau_{max} - \tau_{min}) * min(1, s/\gamma)$ is the annealing coefficient, $s$ is the index of the current training step and $\gamma$ is the number of warm-up steps. This scheme imposes fewer penalties at the beginning and stronger penalties in later stages. As shown in Table 4, using a fixed $\beta = 0.5$ yields competitive performance, but incorporating a learnable prior with a linear schedule $\tau(0.2, 0.5)$ improves results by better balancing compression and reconstruction. This demonstrates that adaptive KL weighting not only stabilizes training but also enables more precise pruning of redundant tokens.

**Limitation.** Our experiments focus on the InternVL family. PGP assumes access to non-compressed visual tokens, whereas architectures such as QwenVL incorporate an explicit token-merging module that performs sequence-level compression and discards fine-grained spatial information. Directly applying PGP to QwenVL is therefore non-trivial; doing so likely requires operating pre-merger, learning a merger-aware posterior, or redesigning the bottleneck to recover spatial structure. We leave this adaptation to future work.

## 5 RELATED WORK

### 5.1 VISION LANGUAGE MODEL

Vision-Language Models (VLMs) have rapidly advanced by aligning visual encoders with large language models, enabling multimodal reasoning across various tasks, including image captioning, visual question answering, and video understanding. Early approaches such as CLIP (Radford et al., 2021) and ALIGN (Jia et al., 2021) demonstrated the power of contrastive pretraining, while more recent instruction-tuned models, including LLaVA (Li et al., 2024a; Sun et al., 2024), QwenVL (Bai et al., 2025; Wang et al., 2024), and InternVL (Chen et al., 2024d;e), leverage lightweight adapters or cross-attention modules to bridge modalities efficiently. These architectures typically concatenate visual tokens from a vision transformer with textual embeddings, enabling joint reasoning but also introducing substantial computational burdens when processing high-resolution images or long videos. The scaling of VLMs toward long-context multimodal understanding has further amplified these challenges, as visual tokens can dominate the sequence length—often exceeding 80% of total tokens. Therefore, reducing the number of visual tokens without compromising semantic fidelity has emerged as a key research direction, motivating recent advances in token pruning and compression.

## 5.2 Visual Token Pruning and Compression

To address the inefficiency of processing redundant visual tokens, a wide range of token compression strategies has been proposed, spanning transformation-based, similarity-based, attention-based, and query-guided methods. Among these, attention-based pruning has attracted particular interest because it directly exploits sparsity in the attention maps of vision transformers or LLMs. Encoder-side methods, such as PruMerge+ (Shang et al., 2024) and VisionZip (Yang et al., 2025), select tokens with high attention relative to the [CLS] token and merge or discard the remainder. Decoder-side methods, including FastV (Chen et al., 2024a) and PyramidDrop (Xing et al., 2025), prune inattentive tokens progressively across layers, guided by the average attention they receive from visual tokens. A line of work employs small-VLM to guide pruning. For instance, SGP (Zhao et al., 2025) utilizes a small model to estimate token relevance by aggregating all the attention weights across all the decoder layers within the decoding stage, reducing the computational overhead of large backbones. While effective, these approaches face practical hurdles when integrated with optimized libraries such as FlashAttention, which obscure explicit attention scores.

## 5.3 Information bottleneck

The Information Bottleneck (IB) (Tishby et al., 2000) formalizes representation learning as a trade-off between task sufficiency and input compression (Dai et al., 2018); its variational form (VIB) (Alemi et al., 2016) makes this trainable at scale by regularizing a parametric posterior toward a simple prior through a KL term while maximizing predictive likelihood. IB/VIB has been widely used for compression and pruning by limiting per-unit information capacity, yielding task-aware sparsity across neurons, channels, and tokens, and often outperforming heuristic saliency measures at aggressive budgets. In Transformers, IB-style regularization has been applied to heads, MLP channels, and layers such as Wang & Yang (2024), as well as token representations in our study, where per-token latent variables are scored via KL-to-prior as a principled importance signal. Complementary "latent bottleneck" modules (e.g., resamplers or token learners) compress vision features into a compact, task-adaptive set (Achille & Soatto, 2018), embodying the same retain-relevant/discard-nuisance principle even when not derived from the IB Lagrangian. Our approach, PGP, instantiates an amortized, token-level IB for VLMs: a small VLM learns $Q_\phi(z|v)$ per visual token; the per-token serves as the importance score.

## 6 Conclusions

We introduced **P**osterior-**G**uided token **P**runing (**PGP**), a principled framework that reframes visual-token pruning in VLMs as amortized variational inference rather than attention- or answer-driven heuristics. By fine-tuning a small-VLM to act as a latent sampler, PGP estimates token importance via the KL divergence between the learned posterior and a prior. This posterior-driven signal is both query- and answer-aware, enabling the small model to detect non-informative tokens while preserving broad regions of informative content for the large model's reasoning. Practically, PGP produces pruning guidance in a single forward pass of the small-VLM, requires no architectural modifications to the large-VLM, and remains fully compatible with optimized kernels such as FlashAttention. Comprehensive experiments demonstrate that PGP achieves state-of-the-art efficiency–accuracy trade-offs across diverse visual tasks. PGP preserves up to 95% of the original accuracy using only 5% of visual tokens, and reduces computational cost by about 40%. These gains stem from replacing brittle attention heuristics with a robust, transferable posterior that better aligns retained tokens with downstream reasoning needs.

**Discussion.** While effective, PGP's amortized posterior relies on a learned small-VLM, leaving room to explore stronger priors. Future work includes extending PGP to multi-image and video inputs, coupling it with retrieval or routing to further reduce sequence length, and analyzing theoretical guarantees for posterior-guided sparsification. We believe PGP offers a scalable and easily deployable route to bring large-VLMs closer to practical, low-latency deployment without sacrificing reasoning quality.

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
