# OpenReview forum: "Posterior-Guided Visual Token Pruning in Vision–Language Models"
_ICLR.cc/2026/Conference — ICLR 2026 Conference Withdrawn Submission_

### Official Review · Reviewer_HTDS · 2025-10-26

**Soundness:** 3
**Presentation:** 3
**Contribution:** 3
**Rating:** 4
**Confidence:** 4

**Summary:**

Vision-language models (VLMs) with dynamic-resolution encoders suffer from inefficiency due to long visual token sequences. This paper proposes Posterior-Guided Pruning (PGP), which introduces a learnable information bottleneck to model the posterior of non-informative tokens conditioned on the query. Tokens with large KL divergence from a learnable prior are retained. PGP produces reliable, query-aware pruning in a single forward pass, preserving broad task-relevant regions. On eight benchmarks, keeping only 5% of visual tokens retains 95% of performance and surpasses prior state-of-the-art by 8%.

**Strengths:**

1. The paper proposes a novel training-based method that is different from attention-based heuristics with a principled variational information bottleneck formulation.

2. The experiments demonstrate an excellent accuracy–ratio trade-offs. With only 5% of visual tokens retained, the large VLM still preserves 95% of its original performance.

3. The paper is very well written and easy to read, with a clear logical flow.

**Weaknesses:**

1. Generalization yet to be verified: The paper lacks experiments on different models; they only conduct experiments primarily on the InternVL family; effectiveness on other architectures (e.g., LLaVA-OV[1], InstructBLIP[2]) remains to be validated.

2. Limited applicability: Assumes the availability of uncompressed visual token sequences, such as QwenVL, and a smaller model of the same architecture; not directly applicable to VLMs with built-in token merging or without a matching small version.

3. Baseline selection is not accurate: The comparison with existing methods is not entirely fair or accurate, as some baselines are not aligned in settings or optimization conditions. For example, the baselines are all training-free methods, which are substantially different from the training setting in this paper. Therefore, more methods should be compared, such as PDrop[3], M3[4], FastVLM[5], and so on.


[1] Bo Li, Yuanhan Zhang, Dong Guo, Renrui Zhang, Feng Li, Hao Zhang, Kaichen Zhang, Yanwei Li, Ziwei Liu, and Chunyuan Li. Llava-onevision: Easy visual task transfer. ArXiv, 2024a.

[2] Wenliang Dai and Junnan Li and Dongxu Li and Anthony Meng Huat Tiong and Junqi Zhao and Weisheng Wang and Boyang Li and Pascale Fung and Steven Hoi. InstructBLIP: Towards General-purpose Vision-Language Models with Instruction Tuning. ArXiv, 2023a.

[3] Long Xing, Qidong Huang, Xiaoyi Dong, Jiajie Lu, Pan Zhang, Yuhang Zang, Yuhang Cao, Conghui He, Jiaqi Wang, Feng Wu, et al. Pyramiddrop: Accelerating your large vision-language models via pyramid visual redundancy reduction. CVPR, 2025.

[4] Cai, Mu and Yang, Jianwei and Gao, Jianfeng and Lee, Yong Jae. M3: Matryoshka Multimodal Models. ICLR, 2025.

[5] Pavan Kumar Anasosalu Vasu, Fartash Faghri, Chun-Liang Li, Cem Koc, Nate True, Albert Antony, Gokul Santhanam, James Gabriel, Peter Grasch, Oncel Tuzel, Hadi Pouransari. FastVLM: Efficient Vision Encoding for Vision Language Models. CVPR, 2025.

**Questions:**

1. Although the paper shows a decrease in TFLOPs, the real runtime benefit is not evident. Could the authors report the actual CUDA time or wall-clock inference latency, considering the extra small-model pass? How much practical speedup is observed?

2. Attention can be noisy due to sink/anchor tokens. I’m curious whether your KL-based posterior scores are more robust than attention-based saliency, or whether similar noise could still leak in via the small VLM. Do you have analyses (e.g., perturbation or correlation tests) to illustrate this?

---

### Official Review · Reviewer_fZ2F · 2025-10-27

**Soundness:** 2
**Presentation:** 2
**Contribution:** 2
**Rating:** 2
**Confidence:** 4

**Summary:**

This paper proposes an efficient visual token pruning method for vision–language models (VLMs). By introducing a learnable information bottleneck in a small VLM, the model identifies non-important visual tokens rather than directly selecting important ones, producing more reliable visual masks. This approach enables the large VLM to retain most of its reasoning capacity while using only a small fraction of tokens. Experiments on eight benchmarks show significant improvements.

**Strengths:**

1. This paper is coherent and well-motivated, effectively demonstrating that identifying non-informative visual tokens via a small VLM can guide efficient pruning. Based on this insight, the authors propose a query-guided, bottleneck-based approach that preserves reasoning performance while substantially reducing the number of visual tokens.

2. The experiments are comprehensive, with the proposed algorithm being validated across multiple benchmarks, demonstrating its feasibility.

**Weaknesses:**

1. I am somewhat skeptical about the practicality of relying on a small VLM in the efficiency domain. The paper only reports the FLOPs reduction of the main model itself, but I would like to see whether the overall system—**including the small model**—**actually achieves a reduction in latency**.

2. As noted by the authors in the Limitations section, PGP cannot currently be adapted to QwenVL, which is a drawback. However, I suggest that the authors try adapting it to models such as LLaVA-Next or LLaVA-OneVision to demonstrate the compatibility of PGP.

3. It would be helpful to include comparisons with methods such as VisionZip and SparseVLM in the table 1.

**Questions:**

1. The comparison in Table 1 seems unfair, as the FastV paper itself recommends pruning in the first two layers. Why, then, is FastV evaluated with pruning at the 9th layer in this comparison?

---

### Official Review · Reviewer_kXKx · 2025-11-03

**Soundness:** 3
**Presentation:** 2
**Contribution:** 2
**Rating:** 4
**Confidence:** 3

**Summary:**

This paper proposes PGP, a posterior-guided visual token pruning framework for vision-language models. Instead of relying on attention or answer-driven heuristics, the method trains a small VLM with a variational information bottleneck to estimate non-informative visual tokens via KL divergence. High-KL tokens are retained for the large model.

Experiments across eight benchmarks show that retaining only 5% of tokens preserves ~95% accuracy, outperforming prior work. The method is efficient, architecture-compatible, and robust on complex queries.

**Strengths:**

1. Well motivated paper: from attention-driven to posterior-driven pruning.

2. Good empirical performance, especially at extreme sparsity.

3. Works with existing architectures and optimized attention kernels.

**Weaknesses:**

1. Primarily evaluated on InternVL; transfer to other VLMs (e.g., Qwen) unclear.


2. Requires training a small VLM module -- not plug-and-play (existing literature).


3.Baseline Coverage is limited.

**Questions:**

1. Generalization Beyond InternVL: applying PGP to models like Qwen-VL or LLaVA-OneVision that compress tokens earlier?

2. What is the additional compute required to train the small VLM bottleneck module? How does this compare to plug-and-play pruning methods in real deployment scenarios?

3. How sensitive is the method to the choice of learnable prior?

4. Several competitive recent pruning methods such as SparseVLM, PyramidDrop, VisionZip, and VScan are not included in your comparison. Could you include these baselines or explain why they were excluded?

5. Can you share examples where PGP fails or under-performs? For example, tasks requiring very localized information or OCR-dense scenes?

---

### Official Review · Reviewer_fH3E · 2025-11-06

**Soundness:** 2
**Presentation:** 3
**Contribution:** 2
**Rating:** 2
**Confidence:** 4

**Summary:**

This work concentrates on the inference acceleration of vLLMs, especially when the input sequence is too long. In particular, basd on existing works of small VLM oriented token pruning, this work detects non-informative visual tokens according to the user’s input query.  By adding a learnable information bottleneck in the small VLM, the method can approximate the posterior distribution of non-important visual tokens, which enables the small model to highlight broad informative regions. Experiments are done on several benchmarks and the results show good performance.

**Strengths:**

The strengths are as follows:
1.Good technical view.  This work formulates token importance estimation as amortized variational inference, providing a sound theoretical basis for pruning guidance.
2.Experiments are done on eight data benchmarks. The results in Sec 4 show promising performance and emperical analysis helps method understanding.

**Weaknesses:**

The weakness are listed as follows:
1.Complexity of the probabilistic model. Since the amortized variational inference formulation may increase model complexity, what is the complexity burden of obtaining this pruning policy? Will the complexity is large than the saved complexity of pruning?
2.Dependence on small-VLM training quality. The proposed PGP is based on small VLMs. So it is important to evaluate the model performance on different small VLM models.
3.Missing related works. There are some othe important works[1,2,3,4] are missing. These works should also be compared and discussed.
4.The experiments are done on specific InternVL model. It is important to verify the method on different architectures such as Qwen, LLaVa.
5.Interpretability of KL-divergence based pruning decisions. Although  KL-divergence-based scoring is principled, there is limited qualitative analysis on what visual patterns are pruned or retained.

[1] Boosting multimodal large language models with visual tokens withdrawal for rapid inference
[2] Dynamic-llava: Efficient multimodal large language models via dynamic vision-language context sparsification.
[3] Visionzip: Longer is better but not necessary in vision language models
[4] Folder: Accelerating multi-modal large language models with enhanced performance

**Questions:**

Se above

---

### Note · Authors · 2025-11-14

**Comment:**

Thank you to all the reviewers for your constructive feedback and the time you dedicated to evaluating our submission. After careful consideration, we have decided to formally withdraw the paper. The reviewers’ comments have been highly valuable, and we will use them to substantially strengthen the manuscript for future submission.

**Withdrawal Confirmation:**

I have read and agree with the venue's withdrawal policy on behalf of myself and my co-authors.